# The Burden of Late Effects and Related Risk Factors in Adolescent and Young Adult Cancer Survivors: A Scoping Review

**DOI:** 10.3390/cancers13194870

**Published:** 2021-09-28

**Authors:** Charlotte Ryder-Burbidge, Ruth L. Diaz, Ronald D. Barr, Sumit Gupta, Paul C. Nathan, Sarah J. McKillop, Miranda M. Fidler-Benaoudia

**Affiliations:** 1Cancer Care Alberta, Alberta Health Services, Holy Cross Centre, Department of Cancer Epidemiology and Prevention Research, 5th Floor, BOX ACB, 2210-2 St. SW, Calgary, AB T2S 3C3, Canada; Charlotte.Ryder-Burbidge@albertahealthservices.ca (C.R.-B.); Ruth.Diaz@albertahealthservices.ca (R.L.D.); 2Health Sciences Centre 3A, Department of Pediatrics, McMaster University, 1280 Main Street West, Hamilton, ON L8S 4K1, Canada; rbarr@mcmaster.ca; 3Department of Pediatrics, Division of Hematology/Oncology, The Hospital for Sick Children, 555 University Avenue, Toronto, ON M5G 1X8, Canada; sumit.gupta@sickkids.ca (S.G.); paul.nathan@sickkids.ca (P.C.N.); 4Department of Pediatrics, Edmonton Clinic Health Academy, University of Alberta, 11405-87 Avenue, Edmonton, AL T6G 1C9, Canada; Sarah.McKillop@albertahealthservices.ca; 5Departments of Oncology and Community Health Sciences, University of Calgary, 2500 University Dr. NW, Calgary, AB T2N 1N4, Canada

**Keywords:** adolescent and young adult, cancer survivors, mortality, chronic disease, second malignant neoplasm

## Abstract

**Simple Summary:**

It is unclear what the risk of negative health outcomes is after cancer during adolescence and young adulthood. We conducted a review to understand the risk of second cancers, chronic conditions, and death in adolescent and young adult (AYA) cancer survivors and found factors that increase the risk. In total, 652 studies were identified, of which 106 were included in the review: 23 for second cancers, 34 for chronic conditions, and 54 for deaths. The number of included studies increased over time, from four studies in 2010 to 17 in 2020. The studies found that AYA cancer survivors are at an increased risk of second cancers, chronic conditions, and deaths. In particular, the following factors increased risk: radiation exposure for second cancers; younger attained age and earlier calendar period of diagnosis for chronic conditions; and non-Hispanic Black or Hispanic, low socioeconomic status, and earlier calendar period of diagnosis for deaths.

**Abstract:**

Risk factors associated with late effects in survivors of adolescent and young adult (AYA) cancer are poorly understood. We conducted a systematic scoping review to identify cohort studies published in English from 2010–2020 that included: (1) cancer survivors who were AYAs (age 15–39 years) at diagnosis and (2) outcomes of subsequent malignant neoplasms (SMNs), chronic conditions, and/or late mortality (>5 years postdiagnosis). There were 652 abstracts identified and, ultimately, 106 unique studies were included, of which 23, 34, and 54 studies related to the risk of SMNs, chronic conditions, and mortality, respectively. Studies investigating late effects among survivors of any primary cancer reported that AYA cancer survivors were at higher risk of SMN, chronic conditions, and all-cause mortality compared to controls. There was an indication that the following factors increased risk: radiation exposure (n = 3) for SMNs; younger attained age (n = 4) and earlier calendar period of diagnosis (n = 3) for chronic conditions; and non-Hispanic Black or Hispanic (n = 5), low socioeconomic status (n = 3), and earlier calendar period of diagnosis (n = 4) for late mortality. More studies including the full AYA age spectrum, treatment data, and results stratified by age, sex, and cancer type are needed to advance knowledge about late effects in AYA cancer survivors.

## 1. Introduction

An estimated 1.2 million adolescents and young adults (AYAs), aged 15–39 years, were diagnosed with cancer in 2020 globally [1,2]. Although the incidence of AYA cancer is increasing or stable in many countries [3], screening, diagnosis, and treatment continue to improve, leading to an overall decrease in all-cause and cancer-related mortality [4]. As a result, there is a growing population of AYA cancer survivors who will spend the majority of their lives at risk of late effects due to their cancer and its treatment. However, the burden of late effects and their associated risk factors in AYA cancer survivors are not well understood.

Although several systematic reviews investigating late effects in survivors of childhood [5,6,7,8] and adult [9,10,11] cancers have been published recently, to our knowledge, only one systematic literature review investigating late effects specifically among AYA cancer survivors has been published in the last decade, and that focused on the 16–29 year age range [12]. Reviews that examine late effects in both childhood and AYA cancers [13,14,15] discuss a striking lack of studies specifically investigating late effects in AYA cancer survivors relative to a robust body of literature for childhood cancer survivors [14]. Given important social and biological differences of AYA cancer, a review of the current evidence on the relationship between tumor-, treatment-, and patient-related risk factors and common late effects in AYA cancer survivors is warranted. 

We thus sought to conduct a systematic scoping review to examine the current state of the literature studying late effects among AYA cancer survivors. Specifically, we aimed to describe the burden of subsequent malignant neoplasms (SMNs), chronic conditions (including hospitalizations and medication prescriptions as surrogates), and late mortality—key late effects, which lead to substantial years of life lost or living with a disability—in AYA cancer survivors and identify risk factors for each late effect. Finally, by summarizing the available literature, we aimed to identify knowledge gaps that will inform future research. 

## 2. Methods

### 2.1. Eligibility Criteria

The three outcomes of interest in this study were (1) SMNs; (2) chronic conditions, which were physical or psychological in nature and diagnosed by a medical professional or for which a hospitalization or pharmacological prescription acted as a surrogate for a diagnosis (e.g., diagnosis of, hospitalization from, or prescription for a disease); and (3) late mortality, defined as a death occurring more than 5 years after the original cancer diagnosis. Studies that presented results for SMNs or chronic conditions were eligible if these conditions developed at any point after diagnosis. Studies were deemed eligible for inclusion in the scoping review if they met the following criteria defined a priori: (1) The study participants included individuals with a history of cancer who were AYAs (age 15–39 years) at the time of diagnosis, including studies not exclusively focused on AYAs (e.g., 0–19 year olds) if age-stratified results, which captured the AYA subgroup, were reported; (2) the study included one or more of the three outcomes defined above; (3) the research was a prospective or retrospective cohort study to allow the examination of the relationship between the AYA cancer exposure and the late effect; and (4) the study was original research published in English in the years 2010–2020. Conference abstracts, theses, reviews, and sources of grey literature were excluded. Case-control studies were not included because they are less suitable for producing evidence of causality and are more prone to bias than cohort studies.

### 2.2. Study Selection

We searched PubMed using three separate searches that aligned with each of the three outcomes of interest (Appendix A). Identified titles and abstracts published between 1 January 2010 and 31 December 2020 were screened independently by two reviewers (CRB and RLD). Disagreements were resolved by discussion until consensus was reached. If consensus could not be reached, a third expert reviewer (MMFB) was consulted. This process was also used to screen the full text of articles. The reference lists of all included articles were then examined independently by CRB and RLD, and additional articles were included in the scoping review until a consensus among the research team was reached. The following information was then abstracted for each included study: primary author, date of publication, country of origin, purpose of the study, data source, outcome ascertainment (medical records, registries, self-report, etc.), study design, type(s) of cancer studied, diagnosis period, age at diagnosis, survival entry date, overall study size or AYA cancer population size within a larger cohort, length of follow-up, key findings, and identified risk factors. Patient- (e.g., demographic, social, and lifestyle), treatment- (e.g., type and year of treatment), and tumor-related (e.g., type and histology) risk factors were all eligible based on their potential to influence the occurrence of late effects. As per the methodology of scoping reviews, we did not appraise the quality or risk of bias of the included studies [16]. Where possible, we abstracted the most-adjusted risk estimates from the results. 

## 3. Results

The three unique searches in PubMed yielded a total of 652 records (SMNs: 199, chronic conditions: 187, late mortality: 266) (Figure 1). Hand-searching of included studies identified an additional 46 eligible studies. Following title and abstract screening, 225 records were deemed potentially relevant to the scoping review and eligible for full-text screening. After full-text screening, a total of 111 studies (SMNs: 23, chronic conditions: 34, late mortality: 54), of which 106 were unique.

All studies in this review had retrospective cohort designs. The number of included studies increased over time, with 44% being published from 2018–2020 (Figure 2). The studies were conducted predominantly in North America (63%) and Europe (29%). Only 45% captured the entire 15–39 years age range, while a further 17% focused on a childhood cohort, which overlapped with the youngest AYAs. Half of the studies made within-group comparisons, while the remainder used an external comparison group, primarily the general population (30%), childhood cancer survivors (12%), or siblings (5%). Sixty (54%) studies included participants from a mixed-cancer population, while the remaining studies examined specific tumors or tumor groups. In equal proportions, participants entered the study at the time of diagnosis (38%) or as 5-year survivors (38%). Finally, the median sample size across the three outcomes was 3053 AYA participants, ranging from 7 to 401,287.

### 3.1. Subsequent Malignant Neoplasms

Studies investigating the development of any kind of SMN among a mixed-cancer population and reporting a single overall estimate for the standardized incidence ratio (SIR) found that AYA cancer survivors were 1.6 (95% confidence interval [95% CI]:1.6–1.6) to 4.3 (95% CI:3.6–5.1) times more likely to experience an SMN relative to that of a primary malignant neoplasm expected in the general population (Table 1) [17,18,19,20]. Absolute excess risk (AER) for developing an SMN in these same cohorts ranged between 15.9 (95% CI:12.1–19.8) and 25.9 (95% CI not reported) per 10,000 person-years at risk. The risk of developing specific types of SMNs ranged widely, with SIRs ranging from 1.1 (95% CI:1.1–1.2) for breast cancer to 2.0 (95% CI:1.9–2.0) for lung cancer and 3.0 (95% CI:2.6–3.5) for meningeal cancer.

For the development of any SMN after a specific first primary cancer diagnosis, SIRs ranged from 1.2 (95% CI:1.1–1.4) after a bladder cancer diagnosis in men [24] to 9.0 (95% CI:3.4–16.5) after a diagnosis of a bone cancer [24,26]. In particular, Hodgkin lymphoma (HL) survivors experienced the highest risk of SMN, which was 3 to 8 times that of the general population, [17,24] with an excess of 0.1 (95% CI not reported) to 111 (95% CI not reported) SMNs per 10,000 person-years [30,32].

#### Risk Factors Associated with Subsequent Malignant Neoplasms

The influence of sex on SMN risk was inconsistent (Figure 3; Appendix A). When mixed cancer populations were assessed, Chao et al. found female sex to be a risk factor (incidence rate ratio [IRR]:1.3, 95% CI:1.1–1.6), [25] while Aben et al. found it to be protective (SIR_male_:3.1, 95% CI:2.7–3.6; SIR_female_:2.0, 95% CI:1.8–2.3) [21]. Investigations of specific primary tumor groups suggested that female sex was a risk factor for SMN in HL survivors (HR:1.8, 95% CI: 1.0–1.3), [32] but protective in melanoma (HR:0.7, 95% CI:0.6–0.8) and thyroid (OR:0.6, 95% CI:0.5–0.7) cancer survivor populations [34,38]. Similarly, older age at diagnosis was found to increase the risk of developing an SMN overall and for melanoma and thyroid cancer survivors, [25,34,38] while other studies have associated younger age at diagnosis or treatment with increased risk of SMNs in HL survivors [24,29,30].

Possible treatment-related risk factors were reported in eight studies. Four studies reported that having radiotherapy elevated the risk of SMNs up to 2.7 times (95% CI:1.0–7.7) that expected, among AYA cancer survivors overall and among specific tumor groups [17,18,25,32], while two studies reported no significant effect of radiotherapy on the risk of SMNs in HL survivors [25,31]. Conversely, a study noted more than a 4-fold (95% CI:3.3–5.4) increase in the risk of SMNs in survivors of HL who received radiotherapy in addition to chemotherapy, relative to those who received chemotherapy only [28]. Treatment with chemotherapy for any primary malignancy increased the risk of SMN (SIR:6.3, 95% CI:4.1–9.4) [18]. Finally, the cumulative incidence of developing an SMN was higher for survivors of acute lymphoblastic leukemia (ALL) or non-HL (NHL) who received hematopoietic transplant compared to those who did not [37,39].

### 3.2. Chronic Conditions

Compared to control groups, AYAs with any primary malignancy had a 38–90% increased risk of first hospitalization (Table 2) [40,41]. In particular, AYA survivors of leukemia, NHL, HL, and central nervous system (CNS), head and neck, and bone tumors were about 4 times (95% CI:3.7–4.8) [42] more likely to develop chronic conditions, as well as more severe chronic conditions that led to hospitalization, relative to the general population or a matched population [40,41,43,44,45,46,47,48,49]. Similarly, a cohort of AYA survivors of leukemia, CNS malignancy, HL, NHL, Wilms tumor, neuroblastoma, soft-tissue sarcoma, and bone cancer had a 4.2-fold (95% CI:3.7–4.8) increased risk for developing one or more severe or disabling health conditions compared to siblings [42].

When specific conditions were assessed, AYA cancer survivors were at an increased risk of nearly all diseases assessed, with a 2- to 3-fold risk of cardiomyopathy, stroke, premature ovarian failure, chronic liver disease, and renal failure reported compared to a non-cancer population [64]. Studies also identified that circulatory and endocrine systems, and mental health, were impacted negatively by cancer and its treatment among AYAs. Specifically, AYA cancer survivors experience a 2.4-fold (95% CI:1.9–2.9) risk of developing cardiovascular disease (CVD) [58] and a 1.3-fold (95% CI:1.3–1.3) risk of hospitalization with CVD relative to the general population [48]. Five-year survivors of cancer diagnosed between 20–34 have a significantly higher risk of cardiomyopathy or heart failure (HR:3.6, 95% CI:2.8–4.6), atherosclerosis, or brain vascular thrombosis (HR:1.7, 95% CI:1.4–2.0), myocardial infarction or cardiac ischemia (HR:1.8, 95% CI:1.5–2.1), and cardiac arrhythmia (HR:1.4, 95% CI:1.2–1.7) [49]. When results were stratified by tumor groups, the 10-year cumulative incidence of CVD was reported to be the highest among survivors of CNS tumors (7.3%, 95% CI:1.6–8.3), ALL (6.9%, 95% CI:5.2–8.7), and acute myeloid leukemia (AML) (6.8%, 95% CI:5.3–8.7) [46]. 95% Hospitalizations for endocrine disorders were also elevated relative to the general population [59]. Finally, one study reported an 80% increased risk of hospitalizations for mental health and psychiatric disorders among AYA survivors of any cancer, [61] which corresponded to 4.5 times (95% CI:3.9–5.3) more purchases of antidepressants compared to siblings [62] and 1.9 times (95% CI:1.0–3.5) the odds of purchases compared to childhood cancer survivors [51].

#### Risk Factors Associated with Chronic Conditions

Patient-related risk factors for the development of a chronic condition included age at diagnosis and treatment, sex, race/ethnicity, and neighborhood socioeconomic status (SES) (Figure 3; Appendix A). Younger age (vs. older age) at any cancer diagnosis or treatment was associated with a greater risk of hospitalization or hospital contact related to cerebrovascular events (SIR_15–19_:3.6, 95% CI:3.0–4.2 vs. SIR_35–39_:1.2, 95% CI:1.2–1.3), [45] and incidence of endocrine disorders (RR_15–19_:4.0, 95% CI:3.4–4.8 vs. RR_30–34_:1.6, 95% CI1.5–1.8), [59] as well as CVD after HL (coronary heart disease: SIR:8.8, 95% CI:6.3–12.3; heart failure: SIR:8.9, 95% CI:25.2–57.4) [66]. Conversely, older age (vs. younger age) at diagnosis was associated with a higher risk of CVD (HR:3.6, 95% CI:2.1–3.2) [46] and diabetes (IRR:2.4, 95% CI:1.7–3.5) [64] after any cancer, and endocrine disorders after cutaneous melanoma (HR:1.4, 95% CI:1.1–1.7) [38]. Compared to their counterparts in the general population, AYA cancer survivors of a younger attained age were hospitalized at higher rates for cardiovascular [48], endocrine [59], some cerebrovascular [45], and for somatic diseases [40], whereas AYA survivors of an older attained age were hospitalized at similar rates to the comparison group. There was no consistent trend in the effect of sex on hospitalizations or the development of chronic conditions. Among survivors of any type of AYA cancer, female sex was associated with a higher risk of any morbidity leading to hospitalization (RR:1.5, 95% CI:1.2–1.9) [47], anxiety or depression (HR:1.6, 95% CI:1.2–2.1) [55,62], endocrine disorders (HR = 1.9, 95% CI:1.6–2.1) [59], and thyroid disorders (IRR:2.1, 95% CI:1.4–3.0) [64]. In other studies, male sex was associated with a higher risk of any hospitalization (HR:1.8, 95% CI:1.6–1.9) [41], cerebrovascular events (SIR_male_:1.5, 95% CI:1.5–1.6) [45], and CVD (HR:1.4, 95% CI:1.2–1.7) [46].

All of the studies that reported on the relationship between chronic conditions and social determinants of health were conducted in the USA. Compared to Whites, identifying as Black (IRR:2.3, 95% CI:1.4–3.5), Hispanic (IRR:2.2, 95% CI:1.6–3.0), or Asian/Pacific Islander (IRR:1.8, 95% CI:1.1–2.8) was associated with an elevated risk of diabetes for all AYA cancer survivors [64] and hospitalization from any cause for HL survivors [67]. Similarly, residing in a low-SES neighborhood (vs. high-SES) was associated with a higher risk of most chronic conditions among survivors of HL [67] and cutaneous melanoma [38], and a 1.6-fold (95% CI:1.3–1.8) risk of CVD among survivors of any type of cancer [46].

Various cancer therapies, including chemotherapy, radiotherapy, and stem cell transplants, were investigated for their role in the development of chronic conditions. In a mixed-cancer cohort, the combination of chemotherapy, radiotherapy, and surgery was associated with an 80% increase in chronic conditions compared to chemotherapy alone (95% CI:1.1–3.1), but no other treatment combinations were associated with significantly more conditions [47]. When assessed individually, receipt of radiotherapy increased the risk of stroke (IRR:3.5, 95% CI:5.9–37.2), diabetes (IRR:1.9, 95% CI:1.3–2.9), and thyroid disorders (IRR:3.1, 95% CI:2.2–4.4) for AYA survivors of any type of cancer [64], as well as diseases of the nervous system for survivors of brain tumors (HR:3.3, 95% CI:1.8–6.2) [68]. Radiotherapy was linked to a 3.4-fold risk of endocrine (total body or chest vs. none) and cardiac conditions (≥15 Gy chest radiation vs. none) for survivors of bone cancers, leukemia, CNS malignancies, HL, NHL, Wilms tumor, neuroblastoma, and soft tissue sarcoma [42]. However, radiotherapy was not associated with the overall risk of health conditions among ALL survivors [39]. Exposure to cytotoxic drugs was associated with an increased risk of CVD among 2-year survivors of any type of cancer (3.5% vs. 2.0% for radiotherapy only) [46] but was not associated with an overall increase in hospital-related morbidity among survivors of HL [32]. Finally, among survivors of NHL, HL, and ALL, stem cell transplants were also associated with an increased overall risk of more than 20 conditions of various organ systems [37,39,67].

### 3.3. Late Mortality

Studies that captured populations of survivors of all types of AYA cancer reported standardized mortality ratios (SMRs) for all-cause mortality ranging from 4.2 (95% CI:4.0–4.3) to 5.9 (95% CI:4.9–6.9; AER:5.3) (Table 3) [18,49]. SMRs for cause-specific deaths were highest for neoplastic causes (i.e., progression or recurrence of primary cancer and second cancers) (SMR:10.9, 95% CI:10.4–11.2 and SMR:7.8, 95% CI:7.0–8.7), infections (SMR:4.0, 95% CI:2.1–5.8), and pulmonary diseases (SMR:7.4, 95% CI:5.7–9.5) [42,49]. Primary diagnoses of ALL (SMR:14.2, 95% CI:7.4–20.9) and cancers of the CNS (SMR:12.3, 95% CI:11.2–13.4 and SMR:7.8, 95% CI:6.6–9.2) were reported as having some of the highest risks of late, all-cause mortality [42,49,70].

#### Risk Factors Associated with Late Mortality

Among AYAs diagnosed with any type of primary cancer, females experienced significantly less excess mortality compared to males when adjusted for years of follow-up, calendar period, and comorbidities (HR:0.7; 95% CI:0.6–0.7) (Figure 3; Appendix A) [70]. One study provided evidence that the conditional survival advantages, specifically for circulatory and respiratory disease deaths, experienced by females for many cancers were most pronounced in the years closer to diagnosis [76]. Females were also reported to have an overall survival advantage relative to males in the following tumor-specific populations: NHL (HR:0.6, 95% CI:0.6–0.8) [83], HL (HR:0.8, 95% CI:0.7–0.9) [31,86], melanoma (HR:0.7, 95% CI:0.6–0.8) [101], AML (HR:0.8, 95% CI:0.7–0.9) [85] and thyroid cancer (HR:0.4, 95% CI:0.3–0.5) [105]. When specific causes of death were assessed, two studies found that males surviving any type of cancer were more at risk of dying from respiratory conditions [77], whereas female survivors of HL and NHL were more at risk of dying from CVD [74]. There was a lack of consensus as to how age at diagnosis affects the risk of late mortality (Figure 3). One study reported significantly more excess mortality (HR:1.4, 95% CI:1.2–1.6) in AYAs diagnosed between the ages of 30–39 compared to AYAs diagnosed between the ages of 15–19 [70]. Conversely, Prasad et al. reported a significantly higher SMR for all causes for AYAs diagnosed between 15–19 (SMR:9.2, 95% CI:7.8–10.6) compared to 20–34 (SMR:5.8, 95% CI:5.4–6.2) [72]. When assessed for specific tumor types, AYAs diagnosed at an older age compared to a younger age also had poorer survival outcomes after NHL [83], HL [86], testicular cancer [106], Ewing sarcoma [108], thyroid cancer [105], and melanoma [101].

When socioeconomic risk factors were investigated, Black AYAs with any type of cancer had a higher risk of death from any cause (HR:1.9, 95% CI:1.8–2.0) [75] overall and specifically after HL [31,86] and testicular cancer [106] compared to non-Hispanic Whites. Non-Hispanic Black (HR:1.7; 95% CI:1.1–2.8) [81] AYA cancer survivors also experienced a greater risk of death from non-cancer causes. Likewise, in Australia, individuals who identified as Aboriginal had a statistically significant 47% increase in excess mortality compared to non-Aboriginals [70]. Similarly, several studies reported higher risks of late mortality for individuals with lower SES or living in lower SES neighborhoods (vs. higher SES) who survived any type of cancer (HR:1.1, 95% CI:1.0–1.3 and HR:1.4, 95% CI:1.3–1.5) [70,78], a brain tumor, HL, leukemia, NHL, thyroid cancer or sarcoma [82], as well as thyroid [105] or testicular cancer [106] and HL [86].

Regarding tumor-related risk factors, having more extensive disease or a higher stage of cancer at diagnosis [81,83,102,107,110] was associated with increased risks in late mortality for tumor-specific and mixed-cancer cohorts. Excluding one study [18], studies reported that excess mortality from AYA cancer decreased over calendar time, with people diagnosed, treated, or enrolled in studies in later years experiencing better survival outcomes [31,32,70,76,80,90,100,106]. For treatment-related risk factors, the evidence associating radiotherapy, chemotherapy, and surgery with late mortality was mixed among the included studies. Two studies conducted in a combined cancer population associated radiotherapy with a 2.0- (95% CI:1.3–3.1) and 1.5- (95% CI:1.0–2.1) fold increase in death from any cause and SMNs, respectively [18,81], while tumor type-specific studies reported non-significant associations [105,110] or decreased risks [31]. Conversely, among survivors of any type of AYA cancer, the risk of late mortality from any cause was not statistically different based on receipt of chemotherapy [18], though specific tumor types, such as bone and soft tissue sarcomas (SMR:3.2, 95% CI:2.3–4.3) [104], were found to have increased risk of late mortality if exposed to chemotherapy. Finally, not having surgery was associated with an increase in cancer-specific death among survivors of head and neck squamous cell carcinoma (HR:1.6; 95% CI:1.2–2.1) [110] but not in combined cancer [18] or bone and soft tissue sarcoma cohorts [104].

## 4. Discussion

For AYA cancer survivors, the development of late effects after cancer can exacerbate the challenges of being a young person, such as balancing social relationships, new careers, and education with limited practical knowledge and financial resources [113,114]. Recognizing the poor understanding of late effects in AYA cancer survivors, this scoping review was conducted to identify published peer-reviewed research that reported on SMNs, chronic conditions, and late mortality risks in this under-researched population. Our findings emphasize the high burden of negative health outcomes experienced by AYA cancer survivors later in life. What remains less clear, based on the evidence from this review, is which factors contribute most to the increased risk of developing late effects, in part due to the lack of detailed treatment exposure investigations and the high level of heterogeneity among the studies. Genetic and lifestyle factors may also modify the mechanisms of established relationships between cancer therapies and late effects [115,116], yet references to factors like obesity, smoking, and physical activity were scarce within the reviewed literature [46].

It is understood generally that cancer treatments, namely, radiotherapy and chemotherapy, are associated with morbidity and premature mortality, and multimodal therapies intensify these risks [47,117,118,119]. Studies included in this review reported that AYAs in mixed-cancer cohorts experience an estimated 2.6-, 1.9-, and 10.4-fold increase in the risk for SMNs, hospitalization from any health condition, and 25-year all-cause mortality, respectively, compared to control groups [25,81,120]. Studies of childhood cancer survivors have shown a dose-response relationship between radiotherapy and the development of secondary sarcomas and breast, thyroid, CNS, and gastrointestinal cancers [22,121,122,123]. Similarly, specific chemotherapeutic drugs have been identified as risk factors for the late effects investigated in this review, with specific thresholds (e.g., high-dose intravenous methotrexate defined as any single dose ≥1000 mg/m^2^) noted by the Children’s Oncology Group Long-Term Follow-Up Guidelines for Survivors of Childhood, Adolescent and Young Adult Cancers [124]. Based on the literature identified in this review, detailed treatment-related risk factors cannot be identified, as few studies include the necessary treatment exposure data to make such assertions. While entities such as the International Late Effects of Childhood Cancer Guideline Harmonization Group are developing guidelines for long-term follow-up for AYA cancer survivors, the evidence used is drawn primarily from studies of childhood cancer survivors [125].

Treatment type is an important predictor of late morbidity, but unmodifiable risk factors may also play a role [117]. Sex-related differences in survival may be related to biological factors such as sex hormones and immune response, behavioral factors such as females’ increased self-awareness of their bodies [126] or tendency towards health-seeking behaviors compared to men [127], and clinical factors such as differences in screening practices (e.g., colorectal cancer) [70]. Age-related differences in the development of late effects may be fundamentally different between younger and older AYAs, as 15–20 year-olds may still be undergoing growth associated with puberty and the rapid proliferation of tissues brought on by sex hormones [117]. These tissues are particularly vulnerable to damage caused by radiation, potentially affecting the maturation of organs and systems associated with teenaged growth spurts (e.g., gonads and musculoskeletal system) [117,128]. In contrast, AYAs aged 20–39 may be exposed to a different set of carcinogenic exposures at work (e.g., asbestos) or in life (e.g., alcohol and smoking) that might alter their risk profile for late effects.

Variable background risks for mortality and chronic conditions also contextualize the findings in this review. Among studies that examined the risk of chronic conditions by attained age, most found that compared to their counterparts in the general population, AYAs of a younger attained age had a heightened risk of developing these late effects than AYAs of an older attained age. This finding may be explained by the background risk of chronic conditions increasing with age, leading to less excess morbidity among older AYA cancer survivors.

### 4.1. Opportunities for Future Research

The identification of effective cancer treatments with fewer short- and long-term side effects is a research priority for AYA cancer survivors [129]. Before interventions can be improved for long-term safety, the risks associated with current treatments must be better understood. Through this scoping review, we can propose several avenues for future research to help elucidate existing knowledge gaps. Broadly, there is a paucity of evidence generated from large, well-characterized cohort studies with lengthy follow-up, which include persons aged 15–39 and report results according to tumor type, sex, and age at diagnosis [130]. The existence of a body of such evidence would allow health professionals to stratify the risk of their patients for late effects and refer high-risk patients for suitable interventions [119]. Detailed treatment information, such as chemotherapy and radiation dosing, is also needed. Unlike childhood cancer survivors, for whom clinical cohorts such as the North American Childhood Cancer Survivor Study [131] and the St. Jude Lifetime Cohort Study [132] exist, few cohorts of AYA cancer survivors include this level of granularity due to the substantially greater number of AYA cancer survivors and the fact that many cohorts are established using population-based cancer registries in which only crude treatment information is recorded [49].

Another area of future investigation relates to SMNs, as it was the least investigated late effect that we explored. SMNs are a well-studied outcome after childhood cancer and a study included in this scoping review found that AYAs surviving any type of cancer have a higher absolute risk of developing an SMN compared to children or older adult survivors [17]. Lung cancer is reported to account for a large proportion of these excess cancers in AYA cancer survivors, highlighting the need to consider the effect of modifiable lifestyle factors, like smoking, as risk factors for late effects [24]. Some estimates suggest that 35% of second cancers developed by adult cancer survivors can be attributed to the adverse effects of alcohol and smoking [133].

Indeed, we identified only one study [46] that studied lifestyle factors such as physical activity, overweight and obesity, alcohol use, or smoking as possible risk factors for late effects in AYA cancer survivors, despite the potential for poor health behaviors to modify the risk of therapy-related complications, with or without interaction with genetics [115,134]. This area is one of active research within adult oncology that has led to changes in clinical practice [115,135] as positive health behaviors have shown some promise for shielding cancer survivors from SMNs, chronic conditions, and premature mortality, both directly and indirectly. Given that AYA cancer survivors have a significantly higher prevalence of unhealthy lifestyle behaviors relative to individuals with no history of cancer [136,137,138], this knowledge gap presents another important opportunity for future work [118].

Finally, there is a dearth of evidence generated from populations in low- and middle-income countries (LMICs), despite these countries bearing the vast majority of cancer deaths [139]. Compared to high-income countries, survival rates in LMICs are estimated to be 11% to 81% lower, meaning that many people will not survive their cancer long enough to develop late effects [140]. Underdiagnoses, misdiagnoses, delayed presentation, and unavailability of treatment or abandonment of therapy contribute to lower rates of survival [141]. Some of these challenges may be corrected by rectifying the general shortage of healthcare professionals who manage patient loads that are often significantly higher than in most high-income countries [141]. Resource constraints further limit access to cancer drugs deemed essential by the World Health Organization [142,143] as well as the capacity for population-based cancer registries [144]. Without adequate cancer-related surveillance, it is difficult to develop effective policy priorities for improving cancer outcomes and health equity in LMICs. AYA oncology should be a priority matter due to the significant social and economic implications of loss of life due to cancer in this age group [145], which accounts for 40% of the world’s population [114].

### 4.2. Limitations

Despite the comprehensive nature of our scoping review, we must recognize several limitations. First, we found substantial heterogeneity across the included studies in terms of cancer types, subgroups, data sources, reference groups, methodology, and analysis. Additionally, we have not examined the quality of the evidence generated from each of the studies because grading of evidence was beyond the scope of this review. As a result, the findings of this review should be interpreted with caution as the included studies have not been assessed according to their individual clinical and methodological contexts. Another limitation of our review is that the search strategy was limited to one language, one database, and 11 years of evidence, which may contribute to selection bias and the lack of studies conducted in LMICs. Generally, English-language restrictions have not been shown to impact the conclusions of systematic reviews [146]. Additionally, we decided a priori to not include “childhood” or “adult” in our search terms, even though the AYA age group can overlap with each of these more general definitions. However, we are confident that through hand-searching of reference lists, we identified most or all childhood and adult survivor cohorts with results that are relevant to this review.

## 5. Conclusions

AYA cancer survivors experience a high level of morbidity many years into their survival. In this comprehensive scoping review, we summarized peer-reviewed literature published since 2010, describing the late-effect burden and examining patient-, tumor-, and treatment-related risk factors associated with late effects after AYA cancer. In doing so, this review highlighted substantial gaps in knowledge about the experience of AYA cancer survivors compared to what is known about children and middle-aged and older adults. Large, observational cohort studies including the full AYA age spectrum, long follow-up, detailed treatment exposure data, and results stratified by tumor type, age at diagnosis, and sex are still needed to help reduce the disease burden and increase quality of life in this unique cancer survivor population.

## Figures and Tables

**Figure 1 cancers-13-04870-f001:**
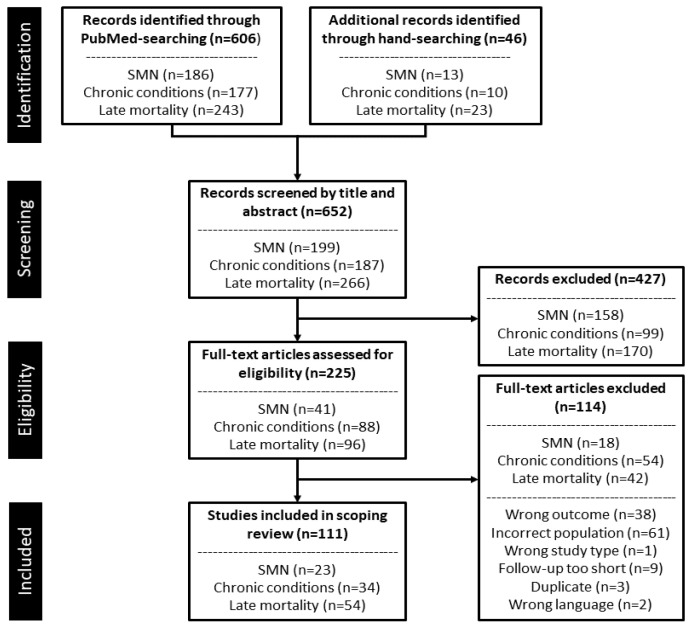
PRISMA flow diagram for the study selection process. Abbreviations: SMN, subsequent malignant neoplasm.

**Figure 2 cancers-13-04870-f002:**
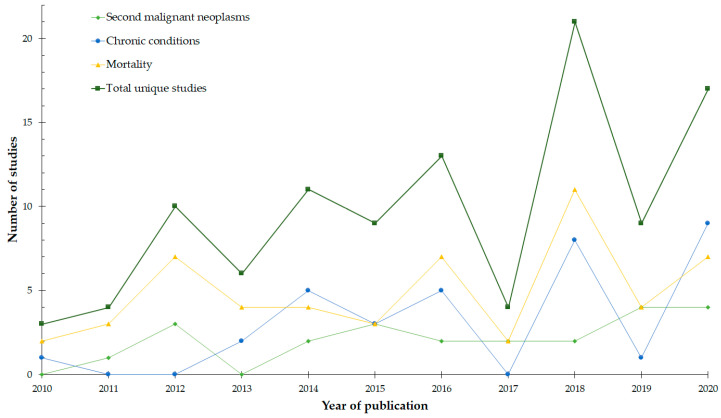
The number of studies included in the scoping review for each outcome of subsequent malignant neoplasms, chronic conditions, and mortality according to year of publication. Duplicates (studies containing information on more than one outcome) were removed from the line indicating the total number of studies.

**Figure 3 cancers-13-04870-f003:**
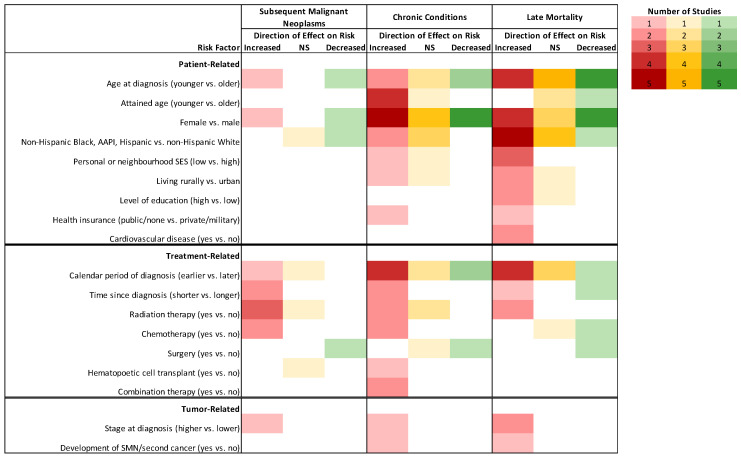
Descriptive representation of the direction of effect for patient-, tumor-, or treatment-related factors described in three or more studies on the risk for subsequent malignant neoplasms, chronic conditions and hospitalizations, and late mortality among AYA cancer survivors in cohorts with combined cancer types. Deeper colors represent more studies (n = 5) and white cells represent that no study investigated that risk factor (n = 0). Reference categories appear in parentheses. Abbreviations: AAPI, Asian-American and Pacific Islander; NS, non-significant; SES, socioeconomic status; SMN, subsequent malignant neoplasm.

**Table 1 cancers-13-04870-t001:** The burden of subsequent malignant neoplasms (SMNs) among adolescent and young adult cancer survivors by tumor group (n = 23).

Reference	Cancer Type	Number of AYA Participants	Outcome Ascertainment	Results
Mixed-Cancer Cohort				
Aben, 2012 [21]	Any primary malignancy except basal carcinomas of the skin	23,161	Netherlands Cancer Registry	At median follow-up time, 1.8% of AYAs developed subsequent cancers.
Henderson, 2012 [22]	First primary malignancy: leukemia, CNS malignancy, HL, NHL, neuroblastoma, soft-tissue sarcoma, kidney cancer or bone cancer. SMN: Sarcoma	2487	Self- or proxy-report questionnaire, and by searches of National Death Index data for US participants	17 out of 2487 (0.7%) AYAs 15–20 developed a gastrointestinal SMN.
Zhang, 2012 [18]	Any primary malignancy	1248	Cohort linked to the population-based Netherlands Cancer Registry (post 1989) or the Dutch Pathology Registry (pre 1989), as well as hospital medical records	62 SMNs were observed. Compared to the general population, SIR for the overall cohort was 3.0 (95% CI 2.3–3.8) with an associated AER of 1.9 per 1000 person-years. SIR for experiencing any type of SMN was 3.6 (95% CI 2.3–5.3) for males and 2.7 (95% CI 1.9–3.7) for females, with associated AERs of 1.9 and 2.0 per 1000 person-years, respectively.
Lee, 2016 [17]	First Primary: leukemia, lymphoma, germ cell tumors (testicular, ovarian), melanoma, thyroid, breast, sarcomas (soft tissue or bone). SMNs: Any cancer with malignant behavior excluding basal cell and cutaneous squamous cell carcinomas	148,558	British Columbia Cancer Registry	7384 patients developed SMN after their original diagnosis. Compared to age- and gender-specific rates, the overall risk of an SMN was 1.6 (95% CI 1.55–1.62) times higher for AYAs (lower than for children, higher than for older adults). AER was 22.9 per 10,000 person-years for AYAs, which was higher than for children or older adults.
Teepen, 2017 [19]	Any primary malignancy	401	SEER	9.6% of those 15-17 years at diagnosis developed a SMN (SIR: 3.3, 95% CI 2.2–4.9; Excess absolute risk [EAR] 25.9 per 10,000 person-years). The SIR for all solid tumors and hematologic malignancies were 3.7 (95% CI 2.4–5.5; EAR 25.9) and 4.1 (95% CI 2.1–7.4; EAR 2.4), respectively, compared to the general population.
Hayek, 2018 [23]	Any primary malignancy	1765	Cohort linked to the Israel National Cancer Registry	75 SPNs were reported in the AYA age group, corresponding with a HR of 1.83 (95% CI 1.21, 2.75).
Bright, 2019 [24]	Breast, cervical, testicular, HL, NHL, melanoma, CNS (intracranial), colorectal, thyroid, soft–tissue sarcoma, ovarian, bladder, other female genital cancers, leukemia, and head and neck	197,827	Office for National Statistics (England) and Welsh Cancer Intelligence and Surveillance Unit, Public Health Wales	12,321 subsequent primary neoplasms were diagnosed in 11,565 survivors, most of whom were survivors of breast cancer, cervical cancer, testicular cancer, and HL.
Chao, 2019 [25]	Any primary malignancy	10,574	Kaiser Permanente Southern California’s SEER-affiliated cancer registry and the California Cancer Registry	622 AYA cancer survivors developed SMN (6.7 per 1000 person-years). Survivors faced 2.6-fold higher risk of developing SMN relative to a comparison cohort.
Fidler, 2018 [26]	First primary malignancy: All cancers. SMN: bone cancers.	11,472	(Varied by country) Population-based cancer registries, late effect clinics, questionnaires, medical records and hospital data, national mortality records, and health insurance registries, validated by pathology or diagnostic reports	Of 11,472 AYA survivors, 10 subsequent primary bone cancers were diagnosed during follow-up time, whereas 1.1 were expected. AYA survivors had an SIR of 9.0 (95% CI 4.3, 16.5) for developing subsequent bone cancers than expected in their age group.
Zakaria, 2019 [20]	Any primary malignancy except epithelial, basal, and squamous skin cancer	7460	Death-linked Canadian Cancer Registry	Among the 15–19 age group, 135 SMNs were observed. Compared to the general population, AYA cancer survivors were 4.3 times as likely (SIR) to experience an SMN (95% CI 3.6–5.1), corresponding with an AER of 15.9 per 10,000 person-years (95% CI 12.1–19.8).
Reulen, 2020 [27]	Any primary malignancy except myelodysplastic syndrome, Langerhans cell histiocytosis, chronic myeloproliferative or lymphoproliferative disorder, or an immunoproliferative disease	21,402	Linkage with population-based national cancer registries, follow-up clinics, questionnaires, available medical records, linkage with national mortality registries and linkage with health insurance registries	Among those diagnosed at age 15–19 years of age, the risk of SPN was: for any digestive (SIR:2.5, 95% CI: 2.1–2.9; AER: 26, 95% CI: 20–34); for colorectal (SIR:1.9, 95% CI: 1.5–2.5; AER: 9, 95% CI: 6–15); for colon only (SIR:2.0, 95% CI: 1.5–2.8; AER:6, 95% CI: 3–11); and for rectum only (SIR:1.8, 95% CI: 1.2–2.6; AER: 3, 95% CI: 1–8); liver (SIR:5.7, 95%, 95% CI: 3.6–8.9; AER: 5, 95% CI: 3–8); stomach (SIR:3.3, 95%, 95% CI: 2.2–4.8; AER: 6, 95% CI: 3–10); and pancreas (SIR:2.3, 95%, 95% CI: 1.4–3.8; AER: 3, 95% CI: 1–6)
Hodgkin lymphoma cohort				
Swerdlow, 2011 [28]	HL	2291	Medical databases, cancer registry information, clinical contact	SMNs developed in 459 of 5798 cohort members.
Swerdlow, 2012 [29]	First Primary: HL. SPN: Breast Cancer	4767	Review of medical records, by responses to questionnaires sent to general practitioners and record linkage with the Netherlands Cancer Registry	Breast cancer or ducta carcinoma in situ developed in 347 AYA cancer survivors. SIR and AERs per 10,000 person-years were elevated in all 5-year age groups.
Schaapveld, 2015 [30]	HL	2736	Case notes, cancer registries, reports from clinicians, screening clinics, and patient reports	For individuals who were aged 15–24 at the time of treatment for first HL, the SIR for experiencing any type of SMN was 8.4 (95% CI 7.5–9.5), with an associated AER of 111 per 10,000 person-years, compared to the general population. For individuals who were aged 25–34 at the time of treatment for first HL, the SIR for experiencing SMN was 5.0 (95% CI 4.4–5.6), with an associated AER of 118 per 10,000 person-years.
Xavier, 2015 [31]	HL	5156	SEER	SMN developed in 122 of 5156 people. At 150 months, the cumulative risk of SMN was 3.3% and 3.0% for people who had and had not received radiation therapy, respectively.
Bhuller, 2016 [32]	First Primary: HL. SPN: All cancers	442	British Columbia Cancer Registry	SIR for developing any SMN: 7.8 (95% CI 5.57–10.52); AER: 5.07 per 1000 person-years. Forty-one survivors (9%) developed SMN; 61% of whom were female. The most frequently developed SMN was breast cancer (n = 14). The risk of developing breast, lung, and thyroid cancer increased the most among cancer survivors.
van Eggermond, 2017 [33]	First primary malignancy: HL. SMN: colorectal cancer	1009	Cohort linkage with a nationwide network and registry of histo- and cytopathology and the Netherlands Cancer Registry	Sixteen cases of colorectal cancer were observed. HL survivors aged 25–34 had an increased risk of developing colorectal cancer compared to the general population (SIR:2.3, 95% CI 1.3–3.7; AER:4.9, 95% CI 1.2–10.3)
Other tumor-specific cohorts				
Goldfarb, 2014 [34]	Thyroid cancer	41,062	National Cancer Database	Among 41,062 cases of thyroid cancer, 1349 (3.3%) had experienced a prior malignancy.
Lee, 2014 [35]	First Primary: osteosarcoma. SMN: all cancers except osteosarcoma	609	SEER	89 participants developed SMN, of whom 16.9% were aged 21–30.
Sultan, 2019 [36]	First primary malignancy: Ewing sarcoma. SMN: All cancers excluding in situ tumors	324	SEER	Of 1131 participants total, 324 were between 20–39 years of age. Of the 324, 9 developed SMN, of whom 8 were aged 20–29 and 1 was aged 30–39.
Abrahao, 2020 [37]	NHL	4392 HIV-uninfected and 425 HIV-infected	California Cancer Registry	Ten-year cumulative incidence of second primary malignancy among HIV-uninfected patients (2·6%, 95% CI 2.0–3.1%) was lower compared to HIV-infected patients (8·1%, 95% CI 5.4–11.4%).
Gingrich, 2020 [38]	Cutaneous melanoma	8259	California Cancer Registry	At 10 years post-diagnosis, 6.4% AYAs developed subsequent cancers. The most common SPN were: melanoma (56.4%), breast (11.8%), thyroid (6.7%), and prostate (2.3%).
Muffly, 2020 [39]	ALL	1069	California Cancer Registry	The 5- and 10-year cumulative incidence of second cancer was 0.4 (95% CI 0.1–1.0) and 1.4 (95% CI 0.7–2.4), respectively.

Abbreviations: AER, absolute excess risk; ALL, acute lymphoblastic leukemia; AYA, adolescent and young adult; CI, confidence interval; CNS, central nervous system; HL, Hodgkin lymphoma; HR, hazard ratio; NHL, non-Hodgkin lymphoma; SEER, National Cancer Institute’s Surveillance, Epidemiology and End Results Program; SIR, standardized incidence ratio; SMN, subsequent malignant neoplasm; SPN, second primary neoplasm.

**Table 2 cancers-13-04870-t002:** The burden of chronic conditions among adolescent and young adult cancer survivors by tumor group (n = 34).

Reference	Cancer Type	Number of AYA Participants	Outcome Ascertainment	Results
Mixed-cancer cohort				
Bradley, 2010 [50]	Any primary malignancy	252	Hospital records	In the 15–19 age group, there were 63 hospitalized survivors and 252 non-hospitalized survivors. OR for risk of hospitalization was 0.69 (95% CI 0.42–1.14) compared to the reference group, which was children aged 0–4.
Deyell, 2013 [51]	Any primary malignancy	1237	PharmaNet, the administrative database that captures all outpatient prescriptions in British Columbia	Adjusted OR of ever using a prescription antidepressant medication among the 15–20 age group was 1.89 (95% CI = 1.04–3.45) and among 20–25 age group was 1.78 95% CI = 0.88–3.5). Reference group was children diagnosed before 5 years.
Zhang, 2014 [47]	Any primary malignancy	902	Hospital records containing morbidity data	455 survivors (50%) had at least one type of late morbidity leading to hospitalization, corresponding to a rate ratio (RR) of 1.37 (95% CI 1.22–1.54) relative to the general population. The highest risks were found for hospitalization due to blood disease (RR = 4.2, 95% CI 1.98–8.78) and neoplasm (RR = 4.3, 95% CI 3.41–5.33).
Brewster, 2014 [52]	Any primary malignancy	3053	National linked database that includes acute hospital discharge records, psychiatric hospital records, and Scottish cancer registration and mortality records	Among people in the AYA age group who were 5-year survivors, the standardized bed day ratio (SBDR) for acute hospitalizations was 3.5 (95% CI 3.4, 3.6) for the 15–19 age group and 2.4 (95% CI 2.4, 2.5) for the 20–24 age group. The SBDR for psychiatric hospitalizations was 0.3 for both the 15–19 and 20–24 age groups 95% CI 0.2–0.3 and 0.3–0.3, respectively, compared to the general population.
Kero, 2014 [49]	Any primary malignancy	9401	Finnish hospital discharge registry	Compared to their siblings, cancer survivors aged 20–34 had a higher risk of cardiovascular events: cardiomyopathy/cardiac insufficiency (HR = 3.6, 95% CI 2.8–4.6), atherosclerosis/brain vascular thrombosis (HR = 1.7, 95% CI 1.4–2.0), myocardial infarction/cardiac ischemia (HR = 1.8, 95% CI 1.5–2.1), and cardiac arrhythmia (HR = 1.4, 95% CI 1.2–1.7).
Kirchoff, 2014 [53]	Any primary malignancy	597	Records from the Utah Department of Health statewide inpatient hospitalization claims data	Among 597 AYA cancer survivors captured in this cohort, 292 did not have a hospitalization during the follow-up and 305 did have a hospitalization during follow-up.
Rugbjerg, 2014 [48]	Any primary malignancy	43,153	Danish Patient Register, containing data on hospital admissions	24.5% of survivors were discharged from the hospital with CVD during follow-up (HRR = 1.30, 95% CI 1.28–1.33). AER was 393 (95% CI 359–427) per 100,000 person-years compared to a cohort of age- and sex-matched subjects. Venous and lymphatic disease was the leading reason for hospitalization (AER = 133 per 100,000 person-years).
van Laar, 2014 [54]	Any primary malignancy except skin carcinomas and melanomas	1880	Hospital admissions data	The rate of hospitalization in the YA cohort was not significantly higher than the general population (HRR = 1.2, 95% CI 0.9–1.5). However, there was a significant increase in the hospitalization rate for pericardial disease (HRR = 4.0, 95% CI 1.8–8.8), cardiomyopathy and heart failure (HRR = 3.8, 95% CI 2.2–6.6), pulmonary heart disease (HRR = 3.5, 95% CI 2.0–6.4), conduction disorders (HRR = 2.0, 95% CI 1.2–3.2), and hypertension (HRR = 1.8, 95% CI 1.3–2.5).
Ahomaki, 2015 [55]	Any First primary Malignant Neoplasm. Excluded those with SMN	9543	Finnish hospital discharge registry	Compared to siblings, YA survivors had higher risk of organic memory/brain disorders (HR = 2.1; 95% CI 1.4–3.1) and mood disorders (HR = 1.3; 95% CI 1.1–1.5). Females had significantly increased risk for neurotic/anxiety disorders (HR = 1.6, 95% CI 1.2–2.1) compared to their siblings, whereas males did not. Radiotherapy did not explain the differences in psychiatric effects.
Asdahl, 2016 [56]	Any primary malignancy	9921	National patient registries containing hospital admissions data	Survivors had 50% excess gastrointestinal or liver diseases compared to the general population (RR = 1.5, 95% CI 1.4–1.6).
Kero, 2016 [57]	Any primary malignancy	2184	Drug Purchase Registry	Higher HR for purchasing anti-hypertensives (HR 1.5, 95% CI 1.3–1.8), diabetes drugs (HR 1.6, 95% CI 1.1–2.2), and lipid-lowering drugs (HR = 1.6, 95% CI 1.0–2.5) in YA cancer survivors compared to siblings. Among specific cancer diagnosis groups, highest HR values for anti-hypertensives were found in YA ALL (HR 4.8, 95% CI 3.1–7.0) and myeloid leukemia (HR 3.4, 95% CI 2.2–5.1) patients. YA ALL patients showed strongest likelihood of purchasing diabetes drugs compared to siblings (HR 3.7, 95% CI 1.2–9.5)
Chao, 2016 [58]	Any primary malignancy	5673	Kaiser Permanente Southern California electronic health records with linkage across clinical databases	For cancer survivors, incidence rate ratio for developing CVD was 2.4 (95% CI 1.9–2.9) compared to patients without cancer. Highest risk in leukemia (IRR = 4.2, 95% CI 1.7–10.3) and breast cancer (IRR = 3.6, 95% CI 2.4–5.5) survivors. Of the three cardiovascular risk factors examined, having diabetes (IRR = 3.2, 95% CI 1.9–5.5) or hypertension (IRR = 3.7, 95% CI 2.4.–5.7) generally imposed a greater risk for CVD than dyslipidemia (IRR = 1.8, 95% CI 1.1–2.9).
Rugbjerg, 2016 [40]	Any primary malignancy except non-melanoma skin cancer	33,555	Danish National Patient Register containing hospital admissions data	53,052 hospitalizations occurred over the follow-up. RR 1.4 (95% CI 1.37–1.39) for survivors compared to controls. The highest risks of hospitalization were for diseases of the blood and blood-forming organs (hospitalization rate ratio [RR] = 2.0, 95% CI 1.87–2.14), infectious and parasitic diseases (RR = 1.69, 95% CI 1.61–1.77), and new malignant neoplasms (RR = 1.63, 95% CI 1.59–1.68). Overall AER was 2803 (95% CI 2712–2893) per 100,000 person-years.
Bright, 2017 [45]	Any primary malignancy	178,962	Hospital Episode Statistics database	2782 AYA cancer survivors were hospitalized for at least one cerebrovascular event—standardized hospitalization ratio (SHR), 1.40 (95% CI 1.3–1.4). AYA cancer survivors are at 2-fold, 1.5-fold, and 1.4-fold risk of cerebral hemorrhage, cerebral infarction, and other cerebrovascular events, respectively.
Jensen, 2018 [59]	Any primary malignancy	32,584	Danish Patient Register, containing data on hospital admissions	6.5% of survivors had at least one hospital contact for an endocrine disease, while 3.8% were expected (RR 1.7 95% CI 1.7–1.871; AER 236.6 per 100,000 person-years). Hospitalization rate ratios (RR) were highest for testicular hypofunction (RR = 75.1, 95% CI 46.0–122.7), ovarian hypofunction (RR = 14.7, 95% CI 8.3–25.9), and pituitary hypofunction (RR = 11.1, 95% CI 8.1–15.3). Leading reasons for hospital contacts: thyroid disease (38%), testicular dysfunction (17%), and diabetes (14%).
Keegan, 2018 [46]	14 first primary AYA cancers	79,176	California Cancer Registry linked to California Office of Statewide Health Planning and Development hospital discharge data	2.8% of survivors developed CVD.
Krawczuk-Rybak, 2018 [60]	Any primary malignancy	197	Self-report data verified by physicians and medical records and entered into an online registry	Of 197 survivors that were 15–18 at diagnosis, organ/system toxicities were most frequent for the skin (38%), male gonads (36%), circulatory system (29%), and female gonads (23%).
Nathan, 2018 [61]	Any primary malignancy	537	Administrative health databases (Registered Persons Database, the Ontario Health Insurance Plan Claims Database, the National Ambulatory Care Reporting System, the Canadian Institutes of Health Information Discharge Abstract Database the Ontario Mental Health Reporting System, and the Ontario Cancer Registry)	In multivariable regression models controlling for age, sex, and income quintile, the relative risk of mental health care visit rates in survivors of AYA (age 15–18) cancer was 1.81 (95% CI 1.2–2.8) relative to the 0–4 age group (*p* = 0.008). In a similar model predicting severe psychiatric events, the relative risk was 0.66 (95% CI 04–1.0; *p* = 0.072).
Ahomaki, 2019 [62]	Any primary malignancy	4598	Drug Purchase Registry	HR for antidepressant purchases was 4.5 (95% CI 3.9–5.3) among AYA cancer survivors compared to siblings.
Smith, 2019 [43]	Any primary malignancy excluding skin carcinomas and melanomas	2627	Hospital Episode Statistics database	Respiratory admission rates were 74% higher in AYA cancer survivors than the general population (Hospital Rate Ratio 1.74, 95% CI 1.6–1.9). For asthma, pneumonia, and chronic lower respiratory disease, admission rates were 49%, 285%, and 266% higher than the general population, respectively.
de Fine Licht, 2019 [44]	Any primary malignancy	11,822	Drug Purchase Registry	Compared to the population-based comparison cohort, AYA cancer survivors had increased risks for hospital contact and prescriptions for diabetes, hyperlipidemia, and hypertension.
Anderson, 2020 [41]	Any primary malignancy	6330	Hospital discharge data from the Utah Department of Health	Higher risk of hospitalization among AYA cancer survivors compared to matched population (HR = 1.9, 95% CI 1.8–2.1). Rate of hospitalizations was also increased among survivors relative to the comparison cohort (RR = 2.05, 95% CI 1.95–2.14).
Bhandari, 2020 [63]	Solid tumors or non-hematologic malignancy	54	Electronic medical records	The risk of acute kidney/chronic kidney disease in AYA was similar to those diagnosed at age younger than 15 years (OR: 1.30, 95% CI: 0.5–3.4)
Chao, 2020 [64]	Any primary malignancy	6778	Kaiser Permanente Southern California electronic health records with linkage across clinical databases	Incidence rate ratio was significantly increased for nearly all comorbidities. IRRs ranged up to 8.3 (95% CI 4.6–14.9) for avascular necrosis. Survivors had a 2- to 3-fold increase for diseases such as cardiomyopathy, stroke, premature ovarian failure, chronic liver disease, and renal failure. Compared to those without cancer, higher percentage of survivors had 2+ comorbidities at 10 years after index date (40% vs. 20% respectively). Adjusted IRR of developing 2+ incident comorbidities: 1.6 (95% CI 1.5–1.8).
Yu, 2020 [65]	Any malignancy	7	Medical records	Among survivors diagnosed at 15–18 years of age, none developed abnormal puberty. Gonadal dysfunction was observed in 2.6% males (1 out of 3), while none was observed among females.
Suh, 2020 [42]	Leukemia, CNS malignancy, HL, NHL, Wilms tumor, Neuroblastoma, Soft-tissue sarcoma, and Bone cancer	4082	Self-report by participants	Early adolescent and YA cancer survivors had HR of 4.2 (95% CI 3.7–4.8) for developing severe and disabling, life-threatening, or fatal health conditions compared to siblings of the same age.
Hodgkin lymphoma cohort				
van Nimwegen, 2015 [66]	HL	1864	Medical records	Compared to the general population, AYA survivors aged 18–24, 25–29, and 30–39 had a 5.4-fold (95% CI 4.5–6.5), 4.1-fold (95% CI 3.3–5.1), and 2.8-fold (95% CI 2.4–3.3) greater risk of developing coronary heart disease (CHD), respectively, and a 18.7-fold (95% CI 14.5–23.6), 10.4-fold (95% CI 7.5–14.2), and 5.7-fold (95% CI 4.4–7.2) greater risk of developing heart failure (HF), respectively.
Keegan, 2018 [67]	HL	5085	California Cancer Registry linked to hospital data from the Office of Statewide Health Planning and Development	39% of AYAs had a hospital admission more than 2 years post-diagnosis. 26% of AYAs had at least one medical condition and 15% had two or more. Ten-year cumulative incidence of disease was highest for endocrine conditions, but estimates varied by race/ethnicity: lowest for non-Hispanic Whites (CI = 12.2, 95% CI 11.0–13.6) and highest for non-Hispanic Blacks (CI = 21.5, 95% CI 16.7–26.7).
Other tumor-specific cohort				
Bhuller, 2016 [32]	First primary malignancy: HL. SMN: Any cancer based on ICDO-3 with behavior code 3 or higher	281	British Columbia Cancer Registry	Survivors had an almost 1.5-fold increased risk of developing morbidity resulting in hospitalization compared to the general population. Higher proportion of survivors experienced two or more types of morbidity resulting in hospitalization compared to controls (26% vs. 15%, respectively). Most common disease groups requiring hospitalization: SMN (n = 45; 16%), digestive disease (n = 38; 14%), injury and poisoning (n = 35; 12%), genitourinary system (n = 28; 10%), circulatory disease (n = 24; 9%), and respiratory disease (n = 22; 8%).
Gunn, 2015 [68]	Brain tumors	315	Finnish Cancer Registry and Hospital Discharge Registry	Compared to siblings, survivors had the most increased risk for diseases of the nervous system (HR = 9.6, 95% CI 6.6–14.0), diseases of the kidney (HR = 5.9, 2.5–14.1), and diseases of the circulatory system (HR = 4.9, 95% CI 2.9–8.1;) and the least increased risk for disorders of vision or hearing loss (HR = 3.6, 95% CI 1.5–8.5), late endocrine diseases (HR = 2.9, 1.1–8.0), and psychiatric disorders (HR = 2.0, 95% CI 1.2–3.2). Cumulative prevalence for most diagnoses remained increased even 20 years after diagnosis.
Abrahao, 2020 [37]	NHL	4392 HIV-uninfected and 425 HIV-infected	California Cancer Registry linked to hospital data from the Office of Statewide Health Planning and Development	Highest 10-year cumulative incidence of disease among HIV-uninfected patients: endocrine (18.5%, 95% CI 17.2–19.9%), cardiovascular (11.7%, 95% CI 10.6–12.8%), respiratory (5.0%, 95% CI 4.3–5.8%), renal (2.2%, 95% CI 1.8–2.8%), and neurologic (2.2%, 95% CI 1.7–2.7%), liver/pancreatic (2.0%, 95% CI 1.5–2.5%), and avascular necrosis (1.2%, 95% CI 0.9–1.7%).
Gingrich, 2020 [38]	Cutaneous melanoma	8259	California Cancer Registry linked to hospital data from the Office of Statewide Health Planning and Development	8.4% of patients had regional disease. The most commonly diagnosed conditions were hematologic disorders (9.1%), cardiac disease (7.7%), and subsequent cancers (6.4%).
Muffly, 2020 [39]	ALL	1069	California Cancer Registry linked to hospital data from the Office of Statewide Health Planning and Development	The 10-year cumulative incidence of late effects was highest for endocrine disease (28.7, 95% CI 25.8–31.6) and cardiac diseases (17.0, 95% CI 14.6–19.5), and lowest for second cancers (1.4, 95% CI 0.7–2.4) and renal disease (3.1, 95% CI 2.1–4.4). All late effects increased over time.
Perisa, 2020 [69]	Ewing Sarcoma	45	Paper and electronic medical records	Treatment-related complications presented in AYA: Neuropathy (87.5%); cardiotoxicity (26.2%); transfections (Median number: 9, 95% CI: 0–72); admissions for fever and neutropenia (median number: 2.95% CI: 0–11). The differences were not significant compared to the pediatric group except for median number of admissions for fever and neutropenia.

Abbreviations: AER, absolute excess risk; ALL, acute lymphoblastic leukemia; AYA, adolescent and young adult; CI, confidence interval; CNS, central nervous system; CVD, cardiovascular disease; HIV, human immunodeficiency virus; HL, Hodgkin lymphoma; HR, hazard ratio; IRR, incidence rate ratio; NHL, non-Hodgkin lymphoma; RR, rate ratio; SBDR, standardized bed day ratio; SEER, National Cancer Institute’s Surveillance, Epidemiology and End Results Program; SHR, standardized hospitalization ratio; SIR, standardized incidence ratio; SMN, subsequent malignant neoplasm; SPN, second primary neoplasm; YA, young adult.

**Table 3 cancers-13-04870-t003:** The burden of late mortality among adolescent and young adult cancer survivors by tumor group (n = 54).

Reference	Cancer Type	Number of Aya Participants	Outcome Ascertainment	Results
Mixed-cancer cohort				
Garwicz S, 2012 [71]	Any primary malignancy	NR	Death certificates and Cause of Death Registers’ files	HR for all-cause mortality was 1.6 (95% CI 1.43–1.80) for survivors aged 15–19 at diagnosis compared to survivors aged 0–4. HR for mortality from first primary was 1.59 (95% CI 1.37–1.84), from second primary was 1.24 (95% CI 0.89–1.72), from non-cancer causes was 1.82 (95% CI 1.44–2.28).
Prasad P, 2012 [72]	Any solid tumor or hematological malignancy	6297	National Population Register, Statistics Finland	SMR for all causes of death: for ages 15–19 (9.2, 95% CI 7.8–10.6) and for ages 20–34 (5.8, 95% CI 5.4–6.2). SMR for death due to circulatory disease: for diagnosis of HL, 8.4 (95% CI 3.1–18.2) for ages 15–19 and 6.5 (95% CI 4.6–8.9) for ages 20–34. For diagnosis of NHL, 21.8 (95% CI 7.1–50.8) for ages 15–19 and 3.3 (95% CI 1.4–6.5) for ages 20–34. For CNS tumor, 1.2 (95% CI 0.03–6.6; non-significant) for ages 15–19 and 3.2 (95% CI 1.3–6.5) for ages 20–34.
Zhang, 2012 [18]	Any primary malignancy	1248	Ministry of Health Vital Statistics Agency, British Columbia Cancer Registry	Among 1248 YA cancer survivors, 11.1% died more than 5 years after diagnosis. The mortality rate was higher than the rate for the general British Columbian population (SMR 5.9, 95% CI 4.9–6.9; AER 5.3).
Haggar F, 2013 [70]	Any primary malignancy	10,266	Western Australia Cancer Registry, Western Australia Mortality Register, Australian National Death Index	Overall 5-year relative survival rates for AYAs diagnosed with any cancer in the most recent diagnostic period (2000–2004) were 0.84 (95% CI 0.82–0.86) in males and 0.86 (95% CI 0.85–0.88) in females.
Kero A, 2014 [49]	Any primary malignancy except carcinoma in situ lesion of the skin	11,417	National Death Certificate files, Statistics Finland	SMR among AYA cancer survivors was 4.2 (95% CI 4.0–4.3) for all causes of death. Cause-specific SMR was highest for infections (SMR = 4.0, 95% CI 2.1–5.8) and cancer (SMR = 10.9, 95% CI 10.4–11.2); lowest for diabetes (SMR: 0.8, 95% CI 0.2–1.4), “external” (SMR = 0.8, 95% CI 0.6–1.1), and alcohol-related (SMR = 0.8, 95% CI 0.6–1.1).
Chao C, 2016 [58]	Any primary malignancy	5673	Kaiser Permanente Southern California’s electronic health records, California state death records, United States Social Security death records	Higher all-cause mortality in cancer survivors who developed CVD compared to survivors without CVD (HR 10.9, 95% CI 8.1–14.8). Compared to those without CVD, survivors who developed CVD had lower 5- (0.67 with CVD vs. 0.92 without CVD) and 10-year (0.55 vs. 0.90) survival after diagnosis.
Henrique L, 2016 [73]	Neoplasia excluding primary tumors in the CNS	889	Sistema de Informações sobre Mortalidade (system database on mortality)	Adjusting for neoplasia and sex: Higher risk of dying for individuals with non-hematological neoplasia (solid tumors) compared with individuals diagnosed with leukemias and lymphomas (HR: 1.47, 95% CI: 1.12–1.93). Compared with individuals diagnosed with leukemias and lymphomas, individuals diagnosed with non-hematological neoplasia had greater risk of death (HR: 1.51, 95% CI: 1.15–1.99).
Henson K, 2016 [74]	Any primary malignancy	200,945	Office of National Statistics in England and the Welsh Cancer Registry, Health and Social Care Information Center	2016 survivors died of cardiac disease. The SMR for all cardiac diseases was 1.4 (95% CI 1.3–1.4). Compared to the general population, higher SMR was observed for survivors of HL (SMR: 3.8; 95% CI, 3.5–4.2), AML (SMR: 2.7; 95% CI, 1.6–4.4), genitourinary cancers other than bladder cancer (SMR: 2.0; 95% CI, 1.6–2.5), NHL (SMR: 1.7, 95% CI, 1.5–2.1), lung cancer (SMR: 1.7; 95% CI, 1.2–2.4), leukemia other than acute myeloid (SMR: 1.6, 95% CI, 1.0–2.4), central nervous system tumor (SMR: 1.4; 95% CI, 1.1–1.6), cervical cancer (SMR:1.3; 95% CI, 1.1–1.5), and breast cancer (SMR: 1.2; 95% CI, 1.1–1.4).
Berkman A, 2017 [75]	Any primary malignancy	135,705	SEER	Survivors of germ cell cancer (HR 2.03, 95% CI 1.66, 2.48), melanoma (HR 1.89, 95% CI 1.14, 3.14), and HL (HR 1.63, 95% CI 1.44, 1.84) had the highest risk at 20 years. For CVD deaths, specifically, Black survivors of AYA leukemias (HR 1.68, 95% CI: 1.06, 2.65), NHL (HR 3.25, 95% CI 1.56, 6.77), thyroid (HR 14.31, 95% CI 3.44, 59.45), melanoma (HR 2.42, 95% CI 1.89, 3.10), and other cancers (HR 2.54, 95% CI 2.13, 3.05) had a higher risk at 20 years.
Anderson C, 2018 [76]	Any primary malignancy except Kaposi sarcoma	205,954	SEER	At 7 years, relative survival of AYA cancer survivors exceeded 95% compared with the general population. Greater relative survival for patients diagnosed in 1988–2009 compared to those diagnosed in 1973–1987. Survival improvements over time were noted for most cancers.
Fidler M, 2018 [77]	Any primary malignancy	200,945	Office for National Statistics and Welsh Cancer Registry, National Death Registration systems	At the end of follow-up, 17% of TYAC survivors had died, of which, 3.2% were due to respiratory causes. Compared to the general population, TYA survivors were more likely to die from a respiratory cause (SMR: 1.7; 95% CI 1.6 to 1.8).
Hayek S, 2018 [23]	Any primary malignancy	1765	Israel national population register	95 deaths were reported in the AYA age group, corresponding with a HR of 1.54 (95% CI 1.13–2.09).
Keegan T, 2018 [46]	14 first primary AYA cancers	79,176	California Cancer Registry and linkages to state and national vital status databases	In total, 2249 of 79,176 patients developed CVD (2.8%). 9285 patients died over the follow-up period (11.7%).
Anderson C, 2019 [78]	Any primary malignancy except Kaposi sarcoma	401,287	SEER	The 10-year cumulative incidence of noncancer-related death after AYA cancer was 2% and 5% among women and men, respectively. The 20-year cumulative incidence of noncancer-related deaths was 4% and 6%, respectively.
Bagnasco F, 2019 [79]	Any primary malignancy	753	National health system registries	Compared to children ages 0–4 at diagnosis, adolescent survivors had a higher risk of death from recurrence (RR-AER = 2.6, 95% CI 1.8–3.8), but not other causes (RR-AER = 1.1, 95% CI 0.6–1.9). SMR for death from all causes except recurrence was 0.59 (95% CI 0.37–0.92) compared to those aged 0–4.
Chao C, 2019 [25]	Any primary malignancy	10,574	SEER	Higher risk of dying in AYA after developing SMN compared to those in comparison group who developed first cancer (HR = 1.90 (95% CI, 1.61–2.24)). AYA cancer survivors’ 5-year overall mortality after SMN diagnosis was 31.9% (128 of 401).
Moke D, 2019 [80]	Any primary malignancy	225,493	SEER	The 7- and 10-year overall survival probability was higher among those diagnosed in 2001–2017 compared to 1988–2000 (78.1% vs. 66.7%) and (75.3% vs. 64.4%), respectively.
Armenian S, 2020 [81]	Any primary malignancy	10,574	SEER	Survival rate of AYA cancer survivors was 78.5% at 25 years after diagnosis, but was at a 10.4-fold increased risk of death compared to noncancer controls (IRR = 10.4, 95% CI 9.7–11.2). Absolute excess risk for death from any cause was 12.7 per 1000 person-years (95% CI, 11.9–13.4 per 1000 person-years). Fifteen years post-diagnosis, incidence of second cancer mortality exceeded the rate of recurrence-related mortality. Lowest long-term survival in breast cancer survivors (25 years: 59.8%) and the highest long-term survival in thyroid cancer survivors (25 years: 95.3%).
Cuglievan B, 2020 [82]	Brain tumor, HL, Leukemia, non-HL, thyroid cancer, sarcomas (bone or soft-tissue)	201	Electronic medical records	Ten-year overall survival for AYAs was about 78%.
Suh E, 2020 [42]	Leukemia, CNS malignancy, HL, NHL, Wilms tumor, neuroblastoma, soft-tissue sarcoma, and bone cancer	5804	United States National Death Index	SMR among all AYA patients for death from all causes was 5.9 (95% CI 5.5–6.2). SMR was 4.8 (95% CI 4.4–5.1) for non-recurrent, health-related causes, 7.8 (95% CI 7.0–8.7) for SMN, 4.4 (95% CI 3.7–5.2) for cardiac causes, 7.4 (95% CI 5.7–9.5) for pulmonary causes, 2.8 (95% CI 2.4–3.2) for other medical causes, 1.1 (95% CI 0.9–1.3) for external causes. Health-related causes of late mortality (SMNs, CVD, pulmonary disease, other median causes) accounted for 52% of deaths among survivors, followed by 36% for recurrence or progression of primary cancer. Cumulative mortality at 30 years was 23% compared to 16% for childhood cancer survivors.
Lymphoma cohort				
Anton–Culver, 2010 [83]	NHL	3489	Death certificates	Overall, 1081 of 3489 people died in the study cohort. The most common causes of death were due to lymphoma-related causes and human immunodeficiency virus.
Castellino, 2011 [84]	HL	1273	United States National Death Index	The HR for risk of death from any cause for the 15–21 age group was 1.1 (95% CI 0.6–2.0), relative to the <10 age group.
Xavier A, 2015 [31]	HL	5156	SEER	5-year survival was better among patients treated with RT relative to those who were not (96.1% vs. 94.6%, respectively, p = 0.002).
Hossain J, 2015 [85]	AML	2290	SEER	The risk of mortality was 30% greater for males compared to females in the 20–24 age group (HR 1.30, 95% CI 1.12–1.52).
Bhuller K, 2016 [32]	First primary malignancy: HL. SMN: Any secondary malignancy	442	Canadian Vital Statistics Agency	60 deaths reported; half of them within 20 years post diagnosis. Standardized mortality ratio for HL survivors was 8.8 (95% CI: 6.7–11.3). Increased risk of death: 18-fold from SMN, 3-fold from non-malignant disease, and 19-fold from circulatory disease. The risk of death remained persistently elevated up to 35 years from diagnosis due to non-relapse mortality.
Keegan T, 2016 [86]	HL	9353	California Cancer Registry, death certificates	Among 9353 patients, 8108 were still alive at the end of follow-up. The highest number of observed deaths was from HL (7.2%), NHL (1.2) and other cancer (1.1%).
Keegan T, 2018 [67]	HL	5085	State and national vital statistics databases	All medical conditions examined in this study reduced overall and HL-specific survival. Respiratory conditions reduced overall survival the most of any condition (HR 6.17, 95% CI 4.5, 8.5).
Patel C, 2018 [87]	HL	511	National Death Index	The 10-, 15-, 20-, and 25-years post-treatment overall survival probabilities were 92.0%, 87.4%, 83.5%, and 75.4%, respectively.
Leukemia cohort				
Goldman, 2010 [88]	Chronic myeloid leukemia	1373	Center for International Blood and Marrow Transplant Research, Bone Marrow Transplant Registry, National Marrow Donor Project	The relative risk of death, treatment failure, or both among those 20–29 and 30–39 years of age at transplantation did not differ from those of patients age <20 years at HCT transplantation
Chen Y, 2012 [89]	Acute promyelocytic leukemia	372	SEER	Ten-year relative survival (RS) was 0.24 (95% CI 0.16–0.33) for the 1975–1990 period, and 0.60 (95% CI 0.50–0.68) for the 1991–1999 period. Ten-year RS for the most recent period (2000–2008) was not reported.
Hunger S, 2012 [90]	ALL	1515	COG ALL clinical trials	Eight percent of adolescent ALL survivors in the cohort died between 5–9.99 years after the start of the study. Less than one percent died at 10 or more years.
Canner J, 2013 [91]	AML	238	Children’s Cancer Group and COG	Overall survival for AYAs 8 years after study entry was approximately 48%, compared to approximately 58% for younger patients (<16 years old).
Woods W, 2014 [92]	AML	517	COG, Cancer and Leukemia Group B, and Southwest Oncology Group trials	Ten-year overall survival was 45.6% and 34% among the COG and CALG/SWOG cohorts, respectively. Ten-year overall survival was higher for patients aged 16–18 compared to aged 19–21 (43% vs. 32%, *p* = 0.034).
Wolfson J, 2018 [93]	ALL, AML	761	Los Angeles County Cancer Surveillance Program	Seven-year survival probabilities for 15–39-year-old ALL survivors were approximately 38% and 56% for patients treated at non-CCC/COG (other) and CCC/COG (Comprehensive Cancer Centers/ COG) facilities, respectively. Seven-year survival probabilities for 15–39 year old AML survivors were approximately 48% and 49% for patients treated at non-CCC/COG (other) and CCC/COG facilities, respectively.
Zheng C, 2018 [94]	Chronic myeloid leukemia	74	Data from Anhui Provincial Hospital	The seven-year overall and leukemia-specific survival for cord blood transplant patients was 55% and 48%, respectively, compared to sibling-allo-HCT, which was 63% and 61%, respectively.
Baron F, 2020 [95]	Primary or secondary AML	661	EORTC/GIMEMA AML-10 trial	No difference in survival by randomization group type (MXR/IDA vs. DNR) for AYA 15–25 (HR: 0.85, 95% CI: 0.56–1.27) or AYA 26–35 (HR: 1.11, 95% CI: 0.78–1.58). Similarly, no difference in survival was found by donor type (i.e., no donor vs. donor) for AYA 15–25 (HR: 0.66, 95% CI: 0.4–1.1) or 26–35 (HR: 0.65, 95% CI: 0.4–1.03)
Venkitachalam R, 2020 [96]	Acute Promyelocytic Leukemia	246	SEER	The 7-year survival probability was approximately 75.6% among AYA 16–20
Melanoma cohort				
Fossa S, 2011 [97]	Testicular cancer	20,411	SEER	Ten-year cumulative testicular cancer-specific mortality rate for seminoma and nonseminoma was 1.4% (95% CI 1.2% to 1.7%) and 6.1% (95% CI 5.7% to 6.7%), respectively. Significantly decreased mortality was observed for participants aged 40 at diagnosis for seminoma HR: 2.0 (95% CI 1.5 to 2.6) and nonseminoma HR: 2.1 (95% CI, 1.7 to 2.6).
Pollack L, 2011 [98]	Melanoma excluding melanoma in situ	13,383	SEER	10-year melanoma-specific survival was 91.9%. AYAs had better 10-year survival probability than those diagnosed at 40–64 (86.7%) or 65+ (77.0%).
Green A, 2012 [99]	Thin melanomas (< = 1.00 mm)	1381	Queensland Registrar of Births, Deaths, and Marriages	Ten-, 15-, and 20-year thin melanoma survival probabilities were 98.5%, 98.2%, and 97.9%, respectively. Better overall survival from thin melanomas for 15–24 years compared to those 45 or older at diagnosis.
Reed K, 2012 [100]	Cutaneous melanoma	256	Medical records	Ten-year overall survival probabilities by decade of diagnosis were approximately 82.7% for 1970–1979, 89.1% for 1980–1989, 94.3% for 1990–199, and 99.7% for 2000–2009.
Gamba C, 2013 [101]	Invasive melanoma of the skin	8853	National Center for Health Statistics	The results reported are for participants in the 1989–1999 diagnosis period. Overall, males were at high risk of mortality compared to females (HR: 1.45, 95% CI 1.25–1.67).
Plym A, 2014 [102]	Invasive malignant melanoma	584	National Population Register	Eight- and 10-year cumulative relative survival was approximately 92.1% and 90.9%, respectively
Other tumor-specific cohort				
Smoll N, 2013 [103]	Chordoma	205	SEER	Relative survival rates for AYAs were 69% (95% CI 60–76), 59% (95% CI 49–68), and 56% (95% CI 44–66) at 10, 15, and 20 years, respectively.
Youn P, 2014 [104]	Bone and soft tissue sarcoma	28,844	SEER	All-cause mortality in survivors was 76% higher compared to that of the general population (SMR 1.76, 95% CI 1.60–1.92; AER 19). At 20 years, this trend persisted (SMR 1.39, 95% CI 1.04–1.82; AER 20).
Keegan T, 2015 [105]	First invasive thyroid carcinoma excluding Hürthle cell carcinomas	16,827	California Cancer Registry (hospital database linkages including the Social Security Administration)	Compared to women of the same age, AYA men were more likely to die from any cause after a diagnosis of thyroid cancer (HR 2.68, 95% CI 2.14–3.34). Higher risk of death for AYAs diagnosed at 30–34 (HR:1.53, 95% CI 1.16–2.01) and 35–39 (HR: 2.01, 95% CI 1.54–2.62) years of age compared to those diagnosed at 15–29 survival than younger AYAs (HR 1.5–2.0)
DeRouen M, 2016 [106]	Testicular cancer	14,249	SEER	Among AYAs with testicular cancer, there were 753, 41, 46, 504, and 14 all-cause deaths among Whites, Blacks, Asian/PI, Hispanic, and Other survivors, respectively. Approximately half of these deaths were due to testicular cancer.
Lau B, 2016 [107]	Thyroid SMN after any primary non-thyroid malignancy	357	SEER	Compared to those diagnosed with a first primary at 0–14 years of age, AYA diagnosed with first primary at age 15–39 had significant lower OS at 10 (AYA: 83.6% vs. Pediatric: 96.4%), 20 (56.3% vs. 88.0%), and 30 (50.9% vs.88.0%) years post-diagnosis.
Novetsky Friedman D, 2017 [108]	Ewing sarcoma	97	Memorial Sloan Kettering institutional cancer registry	HR for all-cause mortality was 3.0 (95% CI 1.4–6.4) for 20–29 year olds and 4.5 (95% CI, 2.0–10.6) for 30–39 year olds, compared to 0–9 year olds.
Bownes L, 2018 [109]	Malignant ovarian germ cell tumors	3125	National Cancer Data Base	Decreased survival was observed for those from a lower income quartile without insurance and with lower education background. The adjusted cumulative survivals at 100 months from diagnosis by education measured as percentage with no high school were approximately 97.9%, 95.2%, 96.1%, and 95.6% among those with > = 21%, 13.0 to 20.9%, 7.0–12.9%, and <7.0% without high school degree, respectively (*p* = 0.017).
Challapalli S, 2018 [110]	Head and neck squamous cell carcinoma	1777	SEER	The survival rate after 8 years of follow-up was 73%.
Chen I, 2018 [111]	Extracranial solid tumors	4128	SEER	Ten- and 20-year overall survivals for AYA patients were, respectively, 42% and 38% for Ewing sarcoma, 30% and 29% for neuroblastoma, 56% and 53% for osteosarcoma, 41% and 39% for rhabdomyosarcoma, and 59% and 57% for Wilms tumor. Compared to pediatric age group (0–15), AYAs are at higher risk of dying from all of the cancers studied except for osteosarcoma
Chu Q, 2020 [112]	Breast Cancer-women stage I to III	1492	Louisiana Tumor Registry	Taking AYA 18–39 years of age as referent category, the overall survival was similar to those 40–49 (HR:1.01, 95%, 95% CI:0.87–1.17) and 50–59 (HR:1.147, 95% CI: 0.99–1.32). However, those diagnosed at ages 60–69 and 70+ had higher risk of mortality than AYA 18–39 (HR: 1.68, 95% CI: 1.46–1.94) vs. HR: 3.93, 95% CI: 3.40–4.53), respectively.
Perisa M, 2020 [69]	Ewing Sarcoma	45	Nationwide Children’s Hospital Columbus, Ohio	Higher risk of mortality in AYA compared to pediatric patients (HR: 3.10, 95% CI:1.45–6.63).Ten-year overall survival was approximately 45% for AYA and 64% for pediatric patients.

Abbreviations: AER, absolute excess risk; ALL, acute lymphoblastic leukemia; AML, acute myelogenous leukemia; AYA, adolescent and young adult; CI, confidence interval; CNS, central nervous system; CVD, cardiovascular disease; HL, Hodgkin lymphoma; HR, hazard ratio; IRR, incidence rate ratio; NHL, non-Hodgkin lymphoma; RR, rate ratio; SBDR, standardized bed day ratio; SEER, National Cancer Institute’s Surveillance, Epidemiology and End Results Program; SHR, standardized hospitalization ratio; SIR, standardized incidence ratio; SMN, subsequent malignant neoplasm; SPN, second primary neoplasm; YA, young adult.

## Data Availability

The data presented in this study are available in this article and Appendix A.

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
