# Peer review of "The Burden of Late Effects and Related Risk Factors in Adolescent and Young Adult Cancer Survivors: A Scoping Review"

_cancers, 2021, doi:10.3390/cancers13194870_

Round 1
Reviewer 1 Report
This is a clear and thorough review of the available literature on the topic. I agree with the authors that this will be most useful for identifying areas requiring further study. TheThe ranges of results from the reviewed studies is very large, and the data nonspecific in terms of most modifiable factors, such as dissed or type of chemotherapy or radiation used. It is encouraging that the outcome is improving over the years between 2010 and 2020.
As the authors point out, it may be difficult to obtain more detailed data from North America unless organizations are funded to do so. I am pleased that there were considerations in the studies wnd the paper of social determinants of health, not only in terms of SES and racial groups, but also in terms of wealth of the home country. Alas, the US may spend more than the other countries, and be more likely to study the outcome, but is unlikely to have the best outcomes. Perhaps we can encourage studies in counties with more equal distribution of wealth and health care.
I had only two small edits. One is the word Finally is repeated in the conclusion. The other is that SNN is not written out in words in the abstract. This latter may be with the assumption that all readers know what these initials mean, or it could be to accommodate word limits in the abstract.
Author Response
Dear Reviewer 1,
Thank you for the opportunity to submit a revised draft of our manuscript titled “The burden of late effects and related risk factors in adolescent and young adult cancer survivors: A scoping review” to Cancers. We appreciate your thoughtful comments regarding the value this paper can add to the field of AYA oncology; in particular, that it highlights inequities in cancer care both within and across countries. We have incorporated your suggestions into the manuscript and highlighted them using track changes.
Comments from Reviewer 1
- Comment 1: I had only two small edits. One is the word Finally is repeated in the conclusion. The other is that SNN is not written out in words in the abstract. This latter may be with the assumption that all readers know what these initials mean, or it could be to accommodate word limits in the abstract.
Response: We have removed the repetition of the word “Finally” in the conclusion on line 464 and written out SMN as subsequent malignant neoplasm in the abstract on line 32.
Sincerely,
Charlotte Ryder-Burbidge (on behalf of all authors)
17 September 2021
Reviewer 2 Report
Cancers-1365340
The burden of late effects and related risk factors in adolescent and young adult cancer survivors: A scoping review
Thank you for the opportunity to review this manuscript. The authors present a comprehensive overview on the burden of late effects and related risk factors. The authors did an excellent job summarizing the evidence and outlining opportunities for future research in this field. Please find a few comments and suggestions for improvement below.
Abstract
- The reported finding that chronic conditions are associated with younger attained age are surprising and are currently not reported in the respective section in the results (3.2.1). As in the general population I would expect chronic conditions to increase with older age. Please clarify. If this finding is based on studies reporting highest risk estimates in younger survivors relative to similarly aged comparison groups, this should be stated accordingly.
- The authors may want to consider adding a definition of late mortality (i.e. >5 years after diagnosis) to the abstract.
- The abstract currently lacks a conclusion specifying the implications of this work, e.g. in regard to directions for future studies.
Methods
Section eligibility criteria
- The restriction to English language and PubMed is a limitation and may have introduced bias (as acknowledged by the authors). The authors may want to elaborate in more detail how this may have affected their findings. To some extent this may e.g. explain the underrepresentation of studies from LMIC’s.
Section study selection
- The authors did not appraise the quality and risk of bias of the included studies. However, for the reader it would be helpful to provide some information on how the outcomes were assessed in the included studies and whether the identified risk factors were from unadjusted or adjusted analyses. Particularly for chronic conditions it would be important to know whether outcomes were extracted from medical records, registries, or were self-reported. This information could be added to the respective tables and added to the discussion section.
- Please define the risk factors included in the review ideally separately for socio-demographic and clinical factors.
Results
- Can the authors elaborate on how many studies included a comparison group? And if yes, what type of comparison group? This information could be added to the respective tables.
Sections subsequent malignant neoplasms, chronic conditions
- The presented information is very comprehensive, but sometimes difficult to follow for the reader. The authors may consider aligning the sequence of the information presented in the text with the order of the studies within the tables to facilitate navigation.
- Can the authors elaborate why stratified findings by cancer type are presented in these sections (e.g. for chronic conditions)? I would rather expect such findings in the section on risk factors.
Sections risk factors
- Figure 2 is very helpful to condense the vast amount of information provided in the supplementary material.
- The authors provide no information on whether the reported risk factors were derived from adjusted or unadjusted analyses. This relates to my concern above regarding the lack of a quality assessment and may partly explain the observed heterogeneity between studies. Can the authors provide some information whether there were differences in the identified risk factors according to type of analysis?
Author Response
Dear Reviewer 2,
Thank you for the opportunity to submit a revised draft of our manuscript titled “The burden of late effects and related risk factors in adolescent and young adult cancer survivors: A scoping review” to Cancers. We appreciate the time and effort you gave to providing us with insightful feedback on this manuscript. We are happy that you found Figure 3 to be a useful visual representation of the supplementary data. In addition to responding to your comments below, we have incorporated most of your suggestions into the manuscript and highlighted them using track changes.
Comments from Reviewer 2
Abstract
- Comment 1: The reported finding that chronic conditions are associated with younger attained age are surprising and are currently not reported in the respective section in the results (3.2.1). As in the general population I would expect chronic conditions to increase with older age. Please clarify. If this finding is based on studies reporting highest risk estimates in younger survivors relative to similarly aged comparison groups, this should be stated accordingly.
Response: We have added a brief description of results concerning attained age in the chronic conditions results section on lines 253-257 and further referenced these findings in the discussion on lines 419-425. In brief, studies included in our review provide evidence that AYA cancer survivors of a lower attained age may experience more risk of developing certain conditions compared to older AYA cancer survivors and relative to the general population. Two possible explanations for this finding may be:- That risk of chronic conditions in the general population is low at young ages and thus the multiplicative excess observed in young AYA cancer survivors is higher than that observed for older AYA cancer survivors where background risks or chronic conditions begin to increase in the general population; and
- Previous studies have shown that risk of late effects in childhood cancer survivors seem to be greater than that observed in AYA cancer survivors. Thus, there may be a dose response relationship with age.
- Comment 2: The authors may want to consider adding a definition of late mortality (i.e. >5 years after diagnosis) to the abstract.
Response: We agree with the reviewer and have added a definition of “late mortality” to the abstract on line 32. - Comment 3: The abstract currently lacks a conclusion specifying the implications of this work, e.g. in regard to directions for future studies.
Response: We agree with the review and have added a brief description of potential future work to the abstract on lines 42-44. Specifically, we have included the following text: “More studies including the full AYA age spectrum, treatment data, and results stratified by age, sex and cancer type are needed to advance knowledge about late effects in AYA cancer survivors.”
Methods
Section eligibility criteria
- Comment 4: The restriction to English language and PubMed is a limitation and may have introduced bias (as acknowledged by the authors). The authors may want to elaborate in more detail how this may have affected their findings. To some extent this may e.g. explain the underrepresentation of studies from LMICs.
Response: We agree with the reviewer and thus have now included a reference to the fact that our English-language restriction may have contributed to the lack of data from LMICs on lines 489-490. However, we do not believe that this language restriction had a substantial impact on other results. There are concerns that studies published in English and languages other than English may differ in terms of quality and the significance of findings (e.g. English language journals less likely to publish non-significant results). However, systematic reviews investigating the impact of English-language restrictions suggest that restrictions may impact the statistical significance of some pooled results in meta-analyses [1], but have little effect on the conclusions of systematic reviews [1, 2].
Dobrescu, A. I., Nussbaumer, S. B., Klerings, I., Wagner, G., Persad, E., Sommer, I., ... & Gartlehner, G. (2021). Restricting evidence syntheses of interventions to English-language publications is a viable methodological shortcut for most medical topics: a systematic review: Excluding English-language publications a valid shortcut. Journal of Clinical Epidemiology.
- Morrison, A., Polisena, J., Husereau, D., Moulton, K., Clark, M., Fiander, M., ... & Rabb, D. (2012). The effect of English-language restriction on systematic review-based meta-analyses: a systematic review of empirical studies. International journal of technology assessment in health care, 28(2), 138-144.
Section study selection
- Comment 5: The authors did not appraise the quality and risk of bias of the included studies. However, for the reader it would be helpful to provide some information on how the outcomes were assessed in the included studies and whether the identified risk factors were from unadjusted or adjusted analyses. Particularly for chronic conditions it would be important to know whether outcomes were extracted from medical records, registries, or were self-reported. This information could be added to the respective tables and added to the discussion section.
Response: We agree with the reviewers comments. In response, we have added a description to the Methods section on lines 115-116 that describe that whether the risk factors were taken from adjusted or unadjusted analyses. To offer further comment, for results of both burden and risk factors and where available, we selected the most adjusted estimates available, which included multivariable-adjusted estimates such as hazard ratio for within-group comparisons and age- and sex-adjusted estimates such as standardized incidence ratio for between-group comparisons. In the limitations section, we acknowledge that differing statistical methodologies should be taken into account when assessing the quality of the included studies.
To help readers appraise the risk of bias for the included studies, we have also added a column to the data tables in the main text titled “Outcome ascertainment” at the reviewer’s suggestion and listed this to the list of information abstracted from the studies on line 107. This column describes how the data for mortality, chronic conditions and second malignant neoplasms were retrieved in the study (e.g. registries, medical records, self-report). Only two studies in the chronic conditions section relied upon participant self-report data, one of which was later confirmed by medical records.
- Comment 6: Please define the risk factors included in the review ideally separately for socio-demographic and clinical factors.
Response: Thank you for this comment. We have added more detail to the description of the types of risk factors that we examined in this study on lines 111-113. Broadly, we did not apply any restrictions on what types of risk factors we were interested in studying as we hoped to capture a comprehensive selection of demographic, treatment/clinical, social and lifestyle factors that may be associated with the risk of late effects in AYA cancer survivors.
Results
- Comment 7: Can the authors elaborate on how many studies included a comparison group? And if yes, what type of comparison group? This information could be added to the respective tables.
Response: Thank you for this suggestion. The results column of the main-text data tables references the comparison group used in each study, if applicable. We now describe the percentage of studies using a comparison group and, if they did, what those groups are, to the results section on lines 131-134. About half of studies employed the use of a comparison group, which included the general population, siblings, childhood cancer survivors, or older cancer survivors. In the limitations section, we acknowledge that differing comparison groups may contribute to the heterogeneity observed between studies.
Sections subsequent malignant neoplasms, chronic conditions
- Comment 8: The presented information is very comprehensive, but sometimes difficult to follow for the reader. The authors may consider aligning the sequence of the information presented in the text with the order of the studies within the tables to facilitate navigation.
Response: We thank the reviewer for the suggestion. We have chosen to leave the studies organized by cancer type, date and author name in the tables, because many studies are referenced more than once throughout the paper (both in the burden and risk factors sections, as well as across outcomes); it is thus not possible for the studies to align always with the text. - Comment 9: Can the authors elaborate why stratified findings by cancer type are presented in these sections (e.g. for chronic conditions)? I would rather expect such findings in the section on risk factors.
Response: We appreciate this comment from the reviewer. The first section, which describes the burden of the outcome of interest, is presented overall for studies that include mixed cancer populations and overall for studies that included only a specific cancer. The reason why studies that include only one cancer are included here is because there is no comparison in these studies to other cancer types, and thus it is not possible to deduce whether the cancer type is a risk factor; instead, in these studies we consider risk factors only to be those identified within the population.
Sections risk factors
- Comment 10: The authors provide no information on whether the reported risk factors were derived from adjusted or unadjusted analyses. This relates to my concern above regarding the lack of a quality assessment and may partly explain the observed heterogeneity between studies. Can the authors provide some information whether there were differences in the identified risk factors according to type of analysis?
Response: Thank you for this comment. We abstracted results from the most adjusted estimates, when available. Although we tried to group more similar studies together when describing them in the text, each finding should be considered within its own context as results will differ by data source, methodology and level of adjustment. For example, studies that use rates to compare late effects in AYA cancer survivors compared to the general population (e.g. standardized mortality ratio and standardized incidence ratio) and studies that compare results between cancer groups (e.g. hazard ratio) will likely produce different results. The comparison groups used in each study (if any) can be ascertained from the results column in the main text. Additional study characteristics are provided in the Supplemental materials, which will further help readers identify differences between the studies and identify potential sources of bias.
Sincerely,
Charlotte Ryder-Burbidge (on behalf of all authors)
17 September 2021